# Blocking NHE Channels Reduces the Ability of In Vitro Capacitated Mammalian Sperm to Respond to Progesterone Stimulus

**DOI:** 10.3390/ijms222312646

**Published:** 2021-11-23

**Authors:** Marc Yeste, Sandra Recuero, Carolina Maside, Albert Salas-Huetos, Sergi Bonet, Elisabeth Pinart

**Affiliations:** 1Biotechnology of Animal and Human Reproduction (TechnoSperm), Institute of Food and Agricultural Technology, University of Girona, ES-17003 Girona, Spain; marc.yeste@udg.edu (M.Y.); sandra.recuero@udg.edu (S.R.); carolina.maside@udg.edu (C.M.); albert.salas@udg.edu (A.S.-H.); sergi.bonet@udg.edu (S.B.); 2Unit of Cell Biology, Department of Biology, Faculty of Sciences, University of Girona, ES-17003 Girona, Spain; 3Department of Nutrition, Harvard T.H. Chan School of Public Health, Harvard University, Boston, MA 02115, USA; 4Consorcio CIBER, M.P., Fisiopatología de la Obesidad y Nutrición (ciBeRobn), Instituto de Salud Carlos III (ISCIII), ES-28029 Madrid, Spain

**Keywords:** sperm, Na^+^/H^+^ exchanger channels (NHEs), in vitro capacitation, acrosome exocytosis

## Abstract

Few data exist about the presence and physiological role of Na+/H+ exchangers (NHEs) in the plasma membrane of mammalian sperm. In addition, the involvement of these channels in the ability of sperm to undergo capacitation and acrosomal reaction has not been investigated in any mammalian species. In the present study, we addressed whether these channels are implicated in these two sperm events using the pig as a model. We also confirmed the presence of NHE1 channels in the plasma membrane of ejaculated sperm by immunofluorescence and immunoblotting. The function of NHE channels during in vitro capacitation was analyzed by incubating sperm samples in capacitating medium for 300 min in the absence or presence of a specific blocker (DMA; 5-(N,N-dimethyl)-amiloride) at different concentrations (1, 5, and 10 µM); acrosome exocytosis was triggered by adding progesterone after 240 min of incubation. Sperm motility and kinematics, integrity of plasma and acrosome membranes, membrane lipid disorder, intracellular calcium and reactive oxygen species (ROS) levels, and mitochondrial membrane potential (MMP) were evaluated after 0, 60, 120, 180, 240, 250, 270, and 300 min of incubation. NHE1 localized in the connecting and terminal pieces of the flagellum and in the equatorial region of the sperm head and was found to have a molecular weight of 75 kDa. During the first 240 min of incubation, i.e., before the addition of progesterone, blocked and control samples did not differ significantly in any of the parameters analyzed. However, from 250 min of incubation, samples treated with DMA showed significant alterations in total motility and the amplitude of lateral head displacement (ALH), acrosomal integrity, membrane lipid disorder, and MMP. In conclusion, while NHE channels are not involved in the sperm ability to undergo capacitation, they could be essential for triggering acrosome exocytosis and hypermotility after progesterone stimulus.

## 1. Introduction

Alkalinization of inner pH (pH_i_) is considered a central event for eliciting the sequence of changes related to sperm capacitation, since it is necessary for further activation of CatSper channels and Ca^2+^ entrance [1,2,3,4]. Different ion channels and transporters are implicated in pH_i_ regulation of sperm cells, being the Na^+^/H^+^ exchangers (NHEs), HCO_3_^−^ membrane transporters, monocarboxylate transporters (MCTs), and voltage gated H^+^ transporters (HVCN1) the most studied in mammals [2,5,6]. Nevertheless, differences between mammalian species exist in the content, pattern of distribution, and physiologic relevance of pH_i_ in the regulation of these channels [5,6,7]. For instance, while HVCN1 channels are not present in the plasma membrane of mouse sperm [8,9], they have been identified in the flagellum of human, pig, and cattle sperm, where they exert a key role in the regulation of hypermotility [2,6,9]. In humans, HVCN1 channels are functionally related to their CatSper counterparts [10], whereas, in pigs, the activity of HVCN1 channels does not seem to be related to Ca^2+^ influx [2], thus suggesting that other H^+^ transporters could participate in the regulation of pH_i_ during sperm capacitation in this species.

NHEs are proton carriers belonging to the family of transporters of solute carrier 9 (SLC9). This family of proteins is composed by different isoforms grouped into three different subfamilies [4,11,12]: (1) NHE1 to NHE5 isoforms, which are localized in the plasma membrane (SLC9A1-5) and NHE6 to NHE9, which are intracellular isoforms (SLC9A6-9); (2) NHA1 and NHA2 isoforms (Na^+^/H^+^ antiporters; SLC9AB1-2); and (3) sNHE (sNHE; SLC9C1, or SLC9A10), which is a sperm-specific plasma membrane isoform identified in mouse and humans. Each NHE isoform is encoded by a specific gene, has a particular characteristic molecular structure and distribution, and a different sensitivity to pharmacological inhibitors [11]. Among plasma membrane isoforms, NHE1 is expressed in nearly all cells [13] and NHE5 is widely distributed, being identified in the brain, spleen, testis, and skeletal muscle [14]; in contrast, NHE2, NHE3, and NHE4 isoforms localize in the digestive system [11,14]. The physiological role of NHA1 and NHA2 is fairly unknown; some evidence indicates that both isoforms are functionally related and highly expressed in renal tubules [15]. Nevertheless, they have also been identified in the testis and sperm flagellum in mice [16].

Few data exist about the type and physiological relevance of NHE isoforms in the plasma membrane of mammalian sperm, most studies having been performed in mouse. In this species, the predominant isoforms are NHE1, NHE5, and sNHE (reviewed in [11]). Whereas NHE1 is essential for pHi regulation and sNHE isoforms regulate sperm motility, the physiological role of NHE5 in mouse sperm fertility is unknown. Interestingly, previous research using patch-clamping in mouse sperm demonstrated that NHE channels are functionally coupled to the CatSper ones [4,11]. Furthermore, recent studies reported that sNHE in humans [17] and NHE1 in sheep [18] are implicated in the regulation of sperm motility. Since not only does the process through which a spermatozoon acquires their capacity to fertilize an oocyte entail changes in sperm kinematics but also in plasma membrane permeability, intracellular calcium levels, mitochondrial activity, and acrosome integrity, the objective of the present study was to determine the physiological relevance of NHE channels in the sperm ability to elicit in vitro capacitation and progesterone-induced acrosome exocytosis. These assays were based on the pharmacological blockage of NHEs channels using a specific inhibitor (5-(N,N-dimethyl)-amiloride; DMA) at different concentrations [4,18]. DMA induces the total blockage of NHE1 and NHE2 isoforms and the partial blockage of NHE5; in contrast, NHE3 and NHE4 are amiloride-resistant isoforms (reviewed in [14]). To the best of our knowledge, no data about the inhibitory effect of this drug on sNHE, NHA1 or NHA2 exist. Considering that only NHE1, NHE5, and sNHE isoforms have been identified in the plasma membrane of mammalian sperm [11,17,18], our pharmacological assay was predicted to lead to the total blockage of NHE1 channels, and either a lack or partial blockage of the other NHE isoforms. At each relevant time point, sperm motility parameters were evaluated using a computer-assisted sperm analysis (CASA) system, whereas plasma membrane and acrosome integrity, mitochondrial activity, and intracellular levels of calcium, superoxide, and overall ROS were measured by flow cytometry. In addition, we analyzed the presence and localization of NHE1 channels in the sperm plasma membrane through immunoblotting and immunofluorescence.

## 2. Results

As this study was conducted using the pig as an animal model and the presence of NHE1 channels was not previously investigated, we first performed immunoblotting and immunofluorescence assays to identify and localize NHE1 channels in the sperm plasma membrane. Following this, the physiological role was determined by incubating sperm samples in capacitation medium in the absence or presence of an NHE-blocking agent (DMA) at 1, 5, or 10 µM for 300 min. In all samples, progesterone was added after 240 min of incubation.

### 2.1. Identification and Immunolocalization of NHE Channels

Immunoblotting assays showed the presence of different bands (from 40 to 120 kDa) in ejaculated sperm samples. In positive controls from pig ovary, oviduct, and epididymis, the bands ranged from 10 to 150 kDa. Peptide competition assays indicated that the 120 kDa-band appearing in sperm samples did not correspond to a NHE1 isoform, whereas that of 75 kDa was specific for NHE1 (Figure 1). Therefore, the molecular weight of NHE1 in pig sperm is 75 kDa.

Immunolocalization assays showed an intense staining in the connecting and terminal pieces of the flagellum, and a weak labeling in the equatorial and acrosomal regions of the head (Figure 2). A diffused immunofluorescence pattern was also observed in the mid-piece which, in some cases, was spread along the principal piece. Peptide competition assays resulted in lack of immunostaining in the equatorial region and flagellum, but not in the acrosomal region. Taken together, these results confirm the specific localization of NHE1 in the equatorial region and in the flagellum.

### 2.2. Sperm Viability and Motility

Sperm viability decreased significantly during the first 120 min of incubation (*p* < 0.05), reaching a value that was maintained without significant variations until the end of the trial in all samples (*p* > 0.05; Figure 3A). During the first 60 min, the decrease in sperm viability was dependent on the DMA concentration, the percentage of viable sperm being significantly higher in the control and samples blocked with 1 μM DMA than in those treated with 5 or 10 μM DMA (*p* < 0.05). Nevertheless, the pattern of variation for sperm viability did not differ significantly between control and blocked samples from 120 min of incubation and until the end of the experiment (*p* > 0.05).

The percentages of total motile/viable sperm (with either a vibrating or linear movement) and of progressively motile/viable sperm (with a mostly straight movement) decreased significantly in the first 60 min of incubation in all samples (*p* < 0.05; Figure 3B,C). The percentage of total motile/viable sperm remained without significant variations until 240 min of incubation (*p* > 0.05), but the addition of progesterone led to a significant increase of this ratio from 250 to 300 min in all samples, despite the pattern of variation differing between treatments (*p* < 0.05). In contrast, the percentage of progressively motile/viable sperm did not differ significantly from 60 to 300 min of incubation in any treatment (*p* > 0.05). Whereas significant differences between control and blocked samples in the percentage of total motile/viable sperm were observed from 250 to 300 min of incubation (*p* < 0.05), that of progressively motile/viable sperm did not differ between treatments at any relevant time point (*p* > 0.05).

### 2.3. Kinematic Parameters

Curvilinear velocity (VCL), straight line velocity (VSL), and average path velocity (VAP) decreased significantly during the first 60 min of incubation (*p* < 0.05), but they were maintained without significant variations until the end of the incubation in both control and blocked samples (*p* > 0.05: Figure 4). A similar pattern of variation was observed for linearity (LIN) and wobble (WOB) parameters (Figure 5A,C), whereas straightness (STR) did not differ throughout the incubation time (*p* > 0.05; Figure 5B).

In control samples, the amplitude of lateral head displacement (ALH) did not differ significantly from 0 to 240 min of incubation (*p* > 0.05), but it increased at 250 min (*p* < 0.05) and then decreased at 270 min (*p* < 0.05; Figure 6A). In all blocked samples, ALH maintained low values throughout the incubation time (*p* < 0.05), regardless of DMA concentration. Beat cross frequency (BCF) decreased progressively throughout the first 180 min (*p* < 0.05), reaching a value that was maintained without significant differences in all treatments (*p* > 0.05; Figure 6B). Comparisons between treatments showed that ALH was the only kinematic parameter differing significantly between the control and blocked samples (*p* < 0.05).

### 2.4. Acrosome Integrity

The percentage of viable sperm with an intact acrosome (PNA-FITC^+^/EthD-1^−^) decreased significantly (*p* < 0.05) and the percentage of viable sperm with an exocytosed acrosome (PNA-FITC^−^/EthD-1^−^) increased (*p* < 0.05) during the first 120 min of incubation (Figure 7). In control samples the percentage of viable sperm with an altered acrosome peaked at 250 min, due to acrosomal exocytosis (*p* > 0.05). In contrast, in blocked samples, both percentages were maintained without significant variations until the end of incubation, regardless of DMA concentration, due to the lack of acrosomal exocytosis.

### 2.5. Membrane Lipid Disorder

The pattern of variation in membrane lipid disorder resembled to that described for acrosome integrity (Figure 8). Therefore, only control samples responded to the progesterone stimulus by significantly increasing the percentage of viable sperm with high membrane lipid disorder (M540^+^/YO-PRO-1^−^) after 250 min of incubation (*p* < 0.05).

### 2.6. Intracellular Calcium Levels

The percentage of viable sperm with low intracellular calcium levels (Fluo3^−^/PI^−^) decreased significantly at 60 min and again at 180 min of incubation (*p* < 0.05), reaching a value that was maintained without significant variations until the end of the experiment (*p* > 0.05; Figure 9A). The percentage of viable sperm with high intracellular calcium levels (Fluo3^+^/PI^−^) and the fluorescence intensity of Fluo3^+^ did not differ between time points in any treatment (Figure 9B,C).

On the other hand, the percentages of viable sperm with low and high (*p* > 0.05) intracellular calcium levels, and the fluorescence intensity of Fluo3^+^ in viable sperm (*p* > 0.05) did not differ between control and blocked samples (*p* > 0.05).

### 2.7. Mitochondrial Membrane Potential

The pattern of variation of mitochondrial membrane potential differed between control and blocked samples (Figure 10A–C). In the control, the percentage of sperm with low mitochondrial membrane potential (LMMP; JC1_agg_^−^/JC1_mon_^+^) decreased (*p* < 0.05) and that of sperm with intermediate mitochondrial membrane potential (MMMP; JC1_agg_^+^/JC1_mon_^+^) increased significantly (*p* < 0.05) during the first 60 min of incubation. Nevertheless, both percentages were maintained without significant variations from 120 to 300 min of incubation (*p* > 0.05). In contrast, the percentage of sperm with high mitochondrial membrane potential (HMMP; JC1_agg_^+^/JC1_mon_^−^) did not differ throughout the incubation period in control samples (*p* > 0.05).

In blocked samples, the percentage of sperm with LMMP decreased significantly during the first 60 min of incubation (*p* < 0.05), whereas those of MMMP and HMMP sperm (*p* < 0.05) increased. However, from this time point and until the end of the experiment, no significant variations in those percentages were observed (*p* > 0.05). Moreover, statistical comparisons between treatments showed significantly lower percentages of LMMP sperm (*p* > 0.05) and significantly higher percentages of MMMP sperm (*p* < 0.05) in the control than in samples blocked with DMA after 300 min of incubation.

The pattern of variation in the JC1_agg_/JC1_mon_ fluorescence ratio of LMMP and MMMP sperm showed no significant changes along incubation in blocked samples, but this ratio increased significantly after 270 min of incubation in the control (*p* < 0.05; Figure 10D,E). The JC1_agg_/JC1_mon_ fluorescence ratio in HMMP sperm was similar between the control and treatments (*p* > 0.05), with two significant increases after 60 min and 270 min of incubation (*p* < 0.05; Figure 10F).

### 2.8. Intracellular ROS Levels

The pattern of variation of the percentage of viable sperm with low ROS levels (DCF^−^/PI^−^) did not differ between control and blocked samples (*p* > 0.05). In all treatments, this percentage decreased progressively during the first 120 min of incubation (*p* < 0.05), reaching a value that was maintained without significant variations until the end of the trial (*p* > 0.05; Figure 11A). The percentage of viable sperm with high ROS levels (DCF^+^/PI^−^) did not differ between treatments (*p* > 0.05), nor did along the incubation time (*p* > 0.05; Figure 11B). Fluorescence intensity of DCF^+^ in DCF^+^/PI^−^ sperm did not differ from 0 to 300 min of incubation in any sample (*p* > 0.05), the fluorescence intensity being statistically similar between treatments (*p* > 0.05; Figure 11C).

The percentage of viable sperm with low (E^−^/YO-PRO-1^−^) and high (E^+^/YO-PRO-1^−^) superoxide levels showed a similar pattern of variation in control and blocked samples (*p* > 0.05; Figure 11D,E). Despite decreasing progressively during the first 120 min of incubation (*p* < 0.05), the percentage of sperm with low superoxide levels did not differ significantly from 180 to 300 min of incubation (*p* > 0.05). In contrast, the percentage of viable sperm with high superoxide levels increased significantly after 60 min and 250 min of incubation with respect to 0 min (*p* < 0.05). The fluorescence intensity of E^+^ in the E^+^/YO-PRO-1^−^ sperm population did not differ between the control and blocked samples, or along the incubation time (*p* > 0.05; Figure 11F).

## 3. Discussion

Few data exist about the types of NHE isoforms present in the plasma membrane of mammalian sperm and their biological significance, most of the studies published thus far being focused on rodents. In addition, and to the best of the authors’ knowledge, no study has previously investigated the involvement of NHE channels in the sperm ability to undergo in vitro capacitation and trigger progesterone-induced acrosome exocytosis. Herein, this issue was addressed using the pig as a mammalian model.

First of all, we confirmed the presence of NHE1 in the plasma membrane of pig sperm, showing a molecular weight of 75 kDa. In a previous study, Garcia and Meizel [19] reported that in pigs the molecular weight of NHE1 was 107 kDa in sperm cells and 110 kDa in erythrocytes. Regarding localization, this channel was observed in the connecting and terminal pieces of the flagellum and the equatorial region of the sperm head. Related with this, it is worth noting that divergences between mammalian species exist about the molecular weight and localization of NHE1. In effect, while NHE1 in most mammalian cell types have a molecular weight ranging from 90 to 120 kDa, these differences being due to variations in the glycosylation pattern [11], NHE1 in sheep sperm has a molecular weight of 85 kDa [18]. Moreover, whilst NHE1 channels are localized in the mid-piece of mouse sperm [20], they are present in the equatorial region, mid-piece, and proximal third of the principal piece of sheep sperm [16]. Such a species-specific distribution throughout the sperm cell is not a characteristic of NHE1, but it is also observed in sNHE [11,17,21] and other ion channels, including SLO3 and HVCN1 (reviewed in [2,22]). Indeed, sNHE channels in mouse sperm have an average molecular weight of 120 kDa and localize in the principal piece [11], whereas in humans they have a molecular weight from 120 to 135 kDa and are present throughout the mid- and principal pieces [17,21]. NHE5 has only been identified in the mid-piece of mouse sperm [20]. Furthermore, the presence of NHE1 in the sperm mid- and principal pieces observed herein for pigs was not homogeneous. The biological significance of these differences is unclear and warrants further research, since although they could be explained by the heterogeneity between ejaculates, the sperm quality of all males included in this study was similar [23]. In agreement with our results, the distribution of sNHE also differs between human ejaculates, such differences being related to the quality of sperm movement [17].

Changes observed in the sperm parameters of control samples during in vitro capacitation agree with previous studies [2,22]. Pharmacological blockage of NHE isoforms with DMA has little impact on in vitro capacitation of pig sperm, as indicated by the lack of differences between control and blocked samples in all sperm parameters analyzed during the first 240 min of incubation. However, this blockage was found to reduce the ability of in vitro capacitated sperm to respond to the progesterone stimulus after 240 min incubation, which manifested in reduced ALH, acrosome exocytosis, and lipid disorder of plasma membrane after 250 min of incubation, and in alterations in total motility and mitochondrial membrane potential from 250 to 300 min of incubation. Related to this, a recent study demonstrated that HVCN1 channels are essential for both in vitro capacitation and progesterone-induced acrosomal exocytosis [2].

Few studies have analyzed the effect of NHE blockage on motility and kinematics of mammalian sperm, but none addressed its relationship to the response to progesterone stimulus. In our approach we used the inhibitor DMA, which blocks totally NHE1 and partially NHE5 isoforms of mammalian sperm [20], its effects on sNHE remaining unknown [4]. In rodents, blockage of NHE channels with DMA results in a decrease of total and progressive motility but not of kinematic parameters [4], whereas in sheep it affects total, but not progressive motility or kinematic parameters including ALH [16]. In contrast, the results obtained in the present study indicate that in pigs DMA blockage has little effect on sperm motility and kinematics during in vitro capacitation, regardless of DMA concentration, although it impairs the motility and mitochondrial membrane potential after progesterone addition to the capacitation medium. Taken together, these results suggest that, in this species, the regulation of sperm movement during in vitro capacitation does not rely on NHE1, but rather on other channels. Interestingly, while NHE1 channels seem to be essential for the sequence of events associated to acrosomal exocytosis after progesterone stimulus, the implication of other NHE channels cannot be ruled out.

Considering the relationship between MMP and progressive motility in ejaculated and capacitated sperm [24,25,26], the changes observed herein in total motility after progesterone addition could be related to the effects of NHE blocking through DMA on mitochondrial function. In effect, samples treated with DMA showed lower percentages of sperm with low and intermediate MMP, and of JC1_agg_/JC1_mon_ fluorescence ratio in both sperm populations. The absence of an effect of NHE blocking on the sperm population with high MMP could be related with the lack of disturbances on progressive motility and intracellular superoxide levels during in vitro capacitation. To our knowledge, this is the first study reporting that the effects of blocking NHE channels rely upon the MMP of sperm cells; further research is necessary to determine its biological relevance. Finally, although progressive motility was not found to be affected by DMA blocking, alterations in ALH were indicative of the inability of blocked samples to switch their motility pattern to hyperactivation after progesterone stimulus [22,27].

In mammals, intracellular alkalinization is currently associated to calcium influx during sperm capacitation, although most of the studies have been performed in mice and humans [9,10,28,29,30]. In rodents, NHE blockage with DMA results in functional alterations not only of CatSper but of also KSper channels [4], whereas in humans HVCN1 channels are functionally related to CatSper [10]. Nevertheless, CatSper channels are also regulated by progesterone, which induces a fast entrance of calcium through CatSper channels with almost no latency in alkaline pH_i_ of sperm cells, although the response to progesterone stimulus differs between mammals (reviewed in [30,31]). Interestingly, in pigs, the blockage of HVCN1 [2] and NHE channels does not affect calcium influx towards the sperm tail in viable sperm, as measured by Fluo3 staining, even after progesterone addition. While the biological significance of these results is unclear, these data suggest that, in this species, CatSper channels are more sensitive to progesterone stimulus than to pH_i_. Nevertheless, further research is necessary to better understand the relationship between intracellular alkalinization and calcium influx in capacitated sperm.

Samples blocked with DMA also showed lower percentages of viable sperm with an altered acrosome and with high membrane lipid disorder after progesterone addition compared to the control, thus suggesting that NHE channels may be involved in acrosomal exocytosis. In contrast to our results, in humans, the blockage of NHE does not impair the ability of capacitated sperm to trigger the acrosomal reaction after progesterone addition [19], nor does blockage affect in vitro fertilization rates in sheep [18]. Despite being essential for human sperm motility and kinematics, sNHE channels have been reported not to be associated to acrosome reaction or calcium influx towards the sperm head [17]. In pigs, NHE are not the only proton channels related to acrosomal exocytosis in capacitated sperm, as HVCN1 channels are also committed [2].

Finally, blocking NHE channels with DMA did not impair sperm viability, which was measured through plasma membrane integrity, during in vitro capacitation. The absence of a deleterious impact on sperm viability could be due to the lack of an effect on the levels of reactive oxygen species (ROS), i.e., hydrogen peroxide (H_2_O_2_) and superoxide (O_2_^−^) levels. Oxidative stress plays an essential role during sperm capacitation by activating the soluble adenylyl cyclase/protein–kinase A pathway (sAC/cAMP/PKA), which induces changes in cell metabolism, tyrosine phosphorylation of flagellar proteins, and efflux of cholesterol from the plasma membrane [26,32]. Nevertheless, increased ROS results in disturbances in plasma membrane integrity and sperm motility, thereby reducing the fertilizing ability of sperm samples [32,33]. On the other hand, few data are available about the functional relationship between pH_i_ alkalinization and ROS production during sperm capacitation. Some studies suggest that HVCN1 channels are involved in the generation of superoxide radicals from the mitochondrial membrane chain in human sperm, despite the underlying regulating mechanism being unknown [24].

## 4. Materials and Methods

### 4.1. Materials

All chemicals were purchased from Sigma-Aldrich Química (Madrid, Spain) unless otherwise indicated.

### 4.2. Semen Samples

The study was performed using the pig as an animal model. Seminal samples (*n* = 9) were collected from nine separate, sexually mature Piétrain boars, aged from 18 to 24 months. All sires were lodged under standard conditions of temperature and humidity, fed a standard diet, and provided with water ad libitum, following the guidelines for animal welfare and handling established by the Animal Welfare Regulations issued from the Regional Government of Catalonia (Barcelona, Spain). Semen samples were purchased from a local farm (Gepork S.A.; Les Masies de Roda, Spain) that operates under standard commercial conditions. Therefore, no specific approval from an ethics committee was required because the authors did not manipulate any animal and the seminal doses purchased were intended for artificial insemination.

Samples were harvested using the gloved-hand technique. The sperm-rich fraction of each ejaculate was immediately filtered through gauze to remove the gel, and diluted 1:1 (*v*:*v*) in a long-term extender (Vitasem, Magapor S.L., Zaragoza, Spain) at 37 °C inside a collecting recipient. Commercial seminal doses were obtained after packaging diluted sperm-rich fractions into 90 mL bags at a concentration of 3 × 10^9^ sperm/dose. Commercial seminal doses were then cooled to 16 °C, and three doses per collection and animal were sent to our laboratory in a heat-insulating container at 16 °C. Once in the laboratory, the sperm quality of all seminal doses was assessed to ensure that all seminal doses included in the study had a sperm motility, viability, and morphology above the previously established thresholds [22].

### 4.3. Experimental Design

In the present study, we first confirmed the presence of NHE1 channels in the plasma membrane of pig sperm, used in this study as a mammalian model, through both immunofluorescence and immunoblotting. Following this, the physiological role of NHE during in vitro capacitation was determined by incubating sperm in capacitating medium in the presence or absence of a specific inhibitor of NHE channels (5-NN-dimethyl amiloride hydrochloride; DMA). The blocking agent was added at the beginning of the experiment (0 min) at a concentration of 1, 5, or 10 μM, in accordance with preliminary experiments and the literature [4,18].

For each independent experiment, seminal doses coming from the same animal were pooled and distributed into 28 aliquots of 9 mL each. Immediately after centrifugation at 600× *g* and 16 °C for 5 min, sperm pellets were resuspended in capacitating medium (20 mM HEPES, 112 mM NaCl, 3.1 mM KCl, 5 mM glucose, 21.7 mM sodium L-lactate, 1 mM sodium pyruvate, 0.3 mM Na_2_HPO_4_, 0.4 mM MgSO_4_·7H_2_O, 4.5 mM CaCl_2_·2H_2_O, 5 mg/mL bovine serum albumin (BSA), and 15 mM sodium bicarbonate) to a final concentration of 1 × 10^7^ sperm/mL. Aliquots were distributed into control samples (seven aliquots) and blocked samples (21 aliquots), the latter being incubated in the presence of DMA at 1, 5, or 10 μM added just before starting the experiment, i.e., 0 min. Samples were incubated at 38.5 °C, 100% humidity, and 5% CO_2_ in a Binder incubator (Binder GmbH, Tuttlingen, Germany) for 60, 120, 180, 240, 250, 270, or 300 min. In samples incubated for 250, 270, and 300 min, 10 µg/mL progesterone was added at 240 min.

At each interval point, the sperm quality was analyzed by assessing sperm motility and kinematics through computer-assisted sperm analysis (CASA, Barcelona, Spain), and sperm viability, acrosome integrity, lipid disorder of plasma membrane, intracellular calcium levels, mitochondrial membrane potential, and intracellular overall ROS and superoxide levels, by flow cytometry. All these sperm parameters were also analyzed just before the beginning of the trial (i.e., 0 min).

### 4.4. Immunoblotting

Immunoblotting assays were conducted following a previously described protocol [19]. Briefly, for total protein extraction, sperm pellets were resuspended in RIPA lysis buffer (R0278), which was supplemented 1:100 (*v*:*v*) with a commercial protease inhibitor cocktail (P8340) containing 0.1 mM phenyl-methane-sulfonylfluoride (PMSF) and 700 mM sodium orthovanadate. Samples were then incubated in agitation at 4 °C for 30 min, sonicated on ice three times (five pulses each; 20 KHz) every 2 min, and centrifuged for 15 min at 10,000× *g* and 4 °C for supernatant collection. Quantification of total protein content in supernatants was carried out in triplicate using a detergent compatible (DC) method (BioRad, Hercules, CA, USA). Once quantified, samples were diluted at 1 µg of total protein per µL in lysis buffer, 10 µL of each sample being mixed with 10 µL of Laemmli reducing buffer 4× containing 5% β-mercaptoethanol (BioRad). Samples were incubated at 95 °C for 5 min and loaded onto 12% polyacrylamide gel (Mini-PROTEAN^®^ TGX™ Precast Gels, BioRad). Gels were run at 20 mA and 120–150 V through an electrophoretic system (IEF Cell Protean System, BioRad). Proteins from gels were then transferred onto polyvinylidene fluoride membranes using Trans-Blot^®^ Turbo™ (BioRad). Subsequently, total protein content was visualized by UV exposition and scanned using a G:BOX Chemi XL system (SynGene, Frederick, MD, USA). Membranes were blocked at room temperature under agitation for 1 h with TBS 1x solution containing 10 mM of Tris (Panreac, Barcelona, Spain), 150 mM of NaCl (LabKem, Barcelona, Spain), 0.05% (*w*:*v*) Tween-20 (pH adjusted to 7.3; Panreac, Barcelona, Spain), and 5% bovine serum albumin (BSA, Roche Diagnostics, S.L.; Basel, Switzerland). Thereafter, membranes were incubated with a primary Na^+^/H^+^ exchanger-1 (NHE1, SLC9A1) antibody (Alomone Labs, Jerusalem, Israel), which was previously diluted in blocking solution at 1:2000 (*v*:*v*) overnight at 4 °C under agitation. In the next step, membranes were rinsed three times with washing solution (TBS1x–Tween20). Following this, membranes were incubated at room temperature under agitation for 1 h with a secondary, antirabbit antibody conjugated with horseradish peroxidase (ref. P0448, Agilent, Santa Clara, CA, USA) diluted at 1:5000 (*v*:*v*) in blocking solution. Membranes were rinsed five times with washing solution and protein bands were visualized with a chemiluminescent substrate (ImmobilionTM Western Detection Reagents, Millipore, Darmstadt, Germany) and scanned with G:BOX Chemi XL 1.4 (SynGene, Cambridge, UK). The specificity of the primary antibody was previously confirmed through a peptide competition assay using the NHE1 blocking peptide (Alomone Labs) at a concentration five times higher than the primary antibody.

### 4.5. Immunofluorescence

Sperm samples were diluted to a final concentration of 5 × 10^6^ cells/mL and washed with PBS (pH = 7.3) at 500× *g* and room temperature for 5 min. Sperm samples were then fixed with 4% (*w*:*v*) paraformaldehyde for 30 min at room temperature, washed twice with PBS at 500× *g* at room temperature for 5 min, and resuspended with PBS. Next, 150 µL of sperm sample was placed onto ethanol-rinsed slides and incubated for 1 h at room temperature to promote sperm adhesion. Adhered sperm were permeabilized by incubating with 1% Triton X-100 in PBS. Afterward, antigen unmasking was performed following the protocol of Kashir et al. [34] with minor modifications. Briefly, sperm on the slides were exposed to acidic Tyrode’s solution for 20 s, and acid was neutralized by washing three times with neutralization solution (Tris 100 mM, pH = 8.5), and three times with PBS.

To block nonspecific binding sites, sperm on the slides were treated with a blocking solution consisting of 5% BSA in PBS at room temperature for 1 h. Subsequently, sperm were first incubated with the primary NHE1 antibody (Alomone Labs; diluted at 1:400, *v*:*v*) at room temperature for 1 h, and then with a secondary antirabbit antibody Alexa Fluor™ Plus 488 (ref. A32731, Invitrogen, Waltham, MA, USA) diluted 1:200 (*v*:*v*) in blocking solution at room temperature for 1 h. Thereafter, samples were washed five times with PBS for 5 min. Finally, samples were mounted with 10 µL of ProLong™ Glass Antifade Mountant with NucBlue™ (Hoechst 33342; ref. P36985, Invitrogen) in the dark. The specificity of the primary antibody was confirmed by separate peptide competition assays; samples were incubated with the NHE1 antibody and its corresponding blocking peptide, which was five times in excess regarding the primary antibody.

Sperm were evaluated under a confocal microscope (CLSM Nikon A1R; Nikon Corp., Tokyo, Japan). Samples were excited at 405 nm to localize the Hoechst 33342-stained nuclei, and at 496 nm to determine the localization of NHE1.

### 4.6. Sperm Motility and Kinematics

Sperm motility was evaluated using a computer-assisted sperm analysis (CASA) system, which consisted of a phase contrast microscope (Olympus BX41; Olympus, Tokyo, Japan) equipped with a video camera and ISAS software (Integrated Sperm Analysis System V1.0; Proiser SL, Valencia, Spain). Three μL of each sample was placed into a prewarmed Leja chamber (IMV Technologies, L’Aigle, France) and observed under a negative phase-contrast field (Olympus 10 × 0.30 PLAN objective). At least 1000 sperm were examined per replicate, and three replicates per sample were evaluated.

For each sperm sample and concentration of inhibitor, we analyzed total (TMOT, %) and progressive sperm motility (PMOT, %), as well as the following sperm kinematic parameters: curvilinear velocity (VCL, μm/s); straight line velocity (VSL, μm/s); average path velocity (VAP, μm/s); amplitude of lateral head displacement (ALH, μm); beat cross frequency (BCF, Hz); linearity (LIN, %), which was calculated assuming that LIN = VSL/VCL × 100; straightness (STR, %), resulting from VSL/VAP × 100; and motility parameter wobble (WOB, %), obtained from VAP/VCL × 100. A sperm cell was classified as motile when its VAP was equal to or higher than 10 μm/s and progressively motile when its STR was equal to or higher than 45%. For each treatment and incubation time, motility parameters were expressed as the mean ± standard error of the mean (SEM; *n* = 9).

### 4.7. Flow Cytometry

Flow cytometry was used to determine sperm viability, acrosome integrity, membrane lipid disorder, intracellular calcium levels, mitochondrial membrane potential (MMP), intracellular levels of superoxide (O_2_^−^) radicals, and intracellular levels of overall ROS. Each sperm parameter was measured by using a proper combination of fluorochromes, all purchased from ThermoFisher Scientific. For proper staining, all samples were diluted to a final concentration of 1 × 10^6^ spermatozoa/mL; after the addition of the corresponding fluorochromes, they were incubated at 38 °C in the dark. A total of three replicates per sample were assessed for each parameter.

Samples were evaluated using a Cell Lab Quanta cytometer (Beckman Coulter, Fullerton, CA, USA). All samples were excited with an argon ion laser (488 nm) set at a power of 22 mW. Cell diameter/volume was assessed employing the Coulter principle for volume assessment, which is based on measuring the changes in electrical resistance produced in an electrolyte solution by suspended, nonconductive particles. In this system, forward scatter (FS) is replaced by electronic volume (EV). Furthermore, 10-μm Flow-Check fluorospheres (Beckman Coulter) were used for EV-channel calibration, by positioning this size of bead at channel 200 on the EV-scale.

Three optical filters were used: FL1 (Dichroic/Splitter, DRLP: 550 nm, BP filter: 525 nm, detection width: 505–545 nm), FL2 (DRLP: 600 nm, BP filter: 575 nm, detection width: 560–590 nm) and FL3 (LP filter: 670 nm/730 nm, detection width: 655–685 nm). FL1 was used to detect green fluorescence from SYBR-14, YO-PRO-1, JC-1 monomers (JC-1_mon_), and 2′,7′-dichlorofluorescein (DCF^+^); FL2 was used to detect orange fluorescence from JC-1 aggregates (JC-1_agg_); and FL3 was used to detect red fluorescence from propidium iodide (PI), merocyanine 540 (M540), and ethidium (E^+^). The signal was logarithmically amplified, and the adjustment of photomultiplier settings was performed according to each staining method.

For all particles, EV and side scatter (SS) were measured and linearly recorded. The sheath flow rate was set at 4.17 μL/min and a minimum of 10,000 events were evaluated per replicate. On the EV channel, the analyzer threshold was adjusted to exclude subcellular debris (particle diameter < 7 μm) and cell aggregates (particle diameter > 12 μm). Therefore, based on EV and SS distributions, the sperm-specific events were positively gated, whereas the others were gated out.

Subsequent data analysis was performed using Flowing Software (Ver. 2.5.1; University of Turku, Turku, Finland) and according to the recommendations of the International Society for Advancement of Cytometry (ISAC). The corresponding mean ± SEM was calculated for each parameter, incubation time, and treatment.

#### 4.7.1. Plasma Membrane Integrity

Sperm viability was evaluated by assessing their plasma membrane integrity using the LIVE/DEAD Sperm Viability Kit (Molecular Probes, Eugene, OR, USA). According to the protocol described by Garner and Johnson [35], sperm were incubated with SYBR-14 (final concentration: 16 nmol/L) and PI (final concentration of 8 μmol/L) for 10 min at 38 °C in darkness. SYBR-14 and PI were excited by the blue laser and their fluorescence was detected through FL1 and FL3 detectors, respectively. According to the staining pattern, three sperm populations in flow cytometry dot plots were obtained: (1) viable, green-stained sperm (SYBR-14^+^/PI^−^); (2) nonviable, red-stained sperm (SYBR-14^−^/PI^+^); (3) nonviable, both red- and green-stained sperm (SYBR-14^+^/PI^+^). A fourth dot population, which corresponded to unstained, nonsperm particles (SYBR-14^−^/PI^−^), was also observed and used to correct data from the other tests. Green-stained spermatozoa (SYBR-14^+^/PI^−^) were used to assess sperm viability. SYBR-14 spill over into the FL3 detector was compensated (2.45%). Results are expressed as percentage of viable spermatozoa (SYBR-14^+^/PI^−^) (mean ± SEM; *n* = 9).

#### 4.7.2. Acrosome Integrity

Acrosomal exocytosis was evaluated by double staining with the lectin peanut agglutinin (PNA; from *Arachis hypogaea*) conjugated with the fluorochrome fluorescein isothiocyanate (FITC) and the vital stain ethidium homodimer (3,8-diamino-5-ethyl-6-phenylphenanthridinium bromide; EthD-1). In brief, samples were incubated with EthD-1 at a final concentration of 2.5 µg/mL and 37.5 °C for 5 min in darkness. Samples were then centrifuged at 2000× *g* and 17 °C for 30 s and resuspended with PBS containing 4 mg/mL bovine serum albumin (BSA). Samples were again centrifuged at the same conditions and then fixed and permeabilized with of 100 µL ice-cold methanol (100%) for 30 s. After methanol removal by centrifugation at 2000× *g* and 16 °C for 30 s, pellets were resuspended with 250 µL PBS. Finally, samples were stained with PNA-FITC (final concentration: 2.5 µM) at 25 °C for 15 min in darkness, washed twice with PBS at 2000× *g* for 30 s and then resuspended in PBS.

After evaluation by flow cytometry, four subpopulations were identified [22]: (1) viable sperm with an intact acrosome (PNA-FITC^+^/EthD-1^−^); (2) viable sperm with an exocytosed acrosome (PNA-FITC^−^/EthD-1^−^); (3) nonviable sperm with an intact acrosome (PNA-FITC^+^/EthD-1^+^); and (4) nonviable sperm with an exocytosed acrosome (PNA-FITC^−^/EthD-1^+^). Fluorescence of EthD-1 was detected through FL3 filter, and that of PNA-FITC through FL1 detector. Results are expressed as the percentage of viable sperm (EthD-1^−^) with either an intact (PNA-FITC^+^) or an exocytosed acrosome (PNA-FITC^−^) (mean ± SEM; *n* = 9).

#### 4.7.3. Plasma Membrane Lipid Disorder

The evaluation of membrane lipid disorder was performed by double-staining with M540 and YO-PRO-1 fluorochromes following a modification of the protocol from Rathi et al. [36] with minor modifications by Yeste et al. [37]. Sperm samples were incubated with M540 at a final concentration of 2.25 μmol/L, and with YO-PRO-1 at a final concentration of 25 nmol/L for 10 min. M540 is a hydrophobic fluorochrome that can intercalate within the membrane. As membrane fluidity increases M540 uptake, this fluorochrome is established as a marker for destabilization of sperm plasma membrane validated in pig, cattle, horse, and dog sperm [38]. M540 fluorescence was detected through the FL3 channel and green fluorescence from YO-PRO-1 was detected through FL1. According to their staining pattern, four populations were identified in flow cytometry dot plots: (1) nonviable sperm with low membrane lipid disorder (M540^−^/YO-PRO-1^+^); (2) nonviable sperm with high membrane lipid disorder (M540^+^/YO-PRO-1^+^); (3) viable sperm with low membrane lipid disorder (M540^−^/YO-PRO-1^−^); and (4) viable sperm with high membrane lipid disorder (M540^+^/YO-PRO-1^−^). The percentage of viable sperm with low membrane lipid disorder (M540^−^/YO-PRO-1^−^) was corrected using the nonsperm particles from the SYBR-14/PI co-staining. Data were not compensated. Results are expressed as the percentage viable sperm with low membrane lipid disorder (M540^−^/YO-PRO-1^−^) and high membrane lipid disorder (M540^+^/YO-PRO-1^−^) (mean ± SEM; *n* = 9).

#### 4.7.4. Mitochondrial Membrane Potential (MMP)

Determination of mitochondrial membrane potential (MMP) was performed with JC-1 following the procedure of Ortega-Ferrusola et al. [39] with minor modifications. Sperm were incubated with 260 nmol/L of JC-1 for 30 min. High MMP causes JC-1 aggregation (JC-1_agg_) and the subsequent emission of orange fluorescence collected through FL2; in contrast, in sperm cells with low MMP, JC-1 remains as a monomer (JC-1_mon_) that emits green fluorescence collected through FL1. Consequently, three different sperm populations can be identified by flow cytometry: (1) sperm with low MMP (green-stained), (2) sperm with high MMP (orange-stained), and (iii) sperm with intermediate MMP (green and orange-stained sperm with heterogeneous mitochondria in the same cell). Data were compensated, as green fluorescence from FL1-channel was subtracted from FL2-channel (51.70%). For each treatment and incubation time, results are expressed as percentages of sperm with low MMP (LMMP), intermediate MMP (MMMP), and high MMP (HMMP) (mean ± SEM; *n* = 9). For each sperm population, the ratio JC1_agg_/JC1_mon_ fluorescence was also determined, the results being expressed as the mean ± SEM (*n* = 9).

#### 4.7.5. Mitochondrial Superoxide (O_2_^−^) Levels

Mitochondrial superoxide (O_2_^−^) radicals were determined by double-staining with hydroethidine (Mito-HE) and YO-PRO-1 fluorochromes [40]. Sperm were incubated with 12 μmol/L HE and 28 nmol/L YO-PRO-1 for 20 min in darkness. HE permeates the sperm plasma membrane and is oxidized to ethidium (E^+^) by O_2_^−^, which produces red fluorescence. In this test, E^+^ fluorescence was detected through FL3; green fluorescence from YO-PRO-1 was detected through FL1. Flow cytometry dot plots allowed the identification of four different sperm populations: nonviable sperm with either low superoxide levels (Mito-E^−^/YO-PRO-1^+^) or high superoxide levels (Mito-E^+^/YO-PRO-1^+^), and viable sperm with either low superoxide levels (Mito-E^−^/YO-PRO-1^−^) and high superoxide levels (Mito-E^+^/YO-PRO-1^−^). YO-PRO-1 spill over into the FL3 detector (7.5%) and Mito-E^+^ spill over into the FL1 detector (7.1%) were compensated. Results are expressed as the percentage viable sperm with low (Mito-E^−^/YO-PRO-1^−^) and high (Mito-E^+^/YO-PRO-1^−^) membrane lipid disorder (mean ± SEM; *n* = 9). The intensity fluorescence of sperm with high mitochondrial superoxide (O_2_^−^) levels was also recorded, and the results were expressed as the mean ± SEM (*n* = 9).

#### 4.7.6. Intracellular Levels of ROS

Intracellular levels of overall ROS were determined with fluorochromes 2′,7′-dichlorodihydrofluorescein diacetate (H_2_DCFDA) and PI, with minor modifications of the procedure described by Guthrie and Welch [40]. Sperm were incubated with H_2_DCFDA (final concentration: 30 μmol/L) for 30 min at 38 °C in the dark. Subsequently, cells were stained with PI (final concentration: 8 µmol/L) for 10 min at 38 °C. H_2_DCFDA is a nonfluorescent probe that, after penetrating the sperm cell membrane, is intracellularly de-esterified and converted into highly fluorescent 2′,7′-dichlorofluorescein (DCF^+^) by ROS. DCF^+^ fluorescence was collected through FL1 as green fluorescence, whereas red fluorescence from PI was detected through FL3. According to their fluorescence emission, four different sperm populations were distinguished in dot plots: viable sperm with either low ROS levels (DCF^−^/PI^−^) or high ROS levels (DCF^+^/PI^−^), and nonviable sperm with either low ROS levels (DCF^−^/PI^+^) or high ROS levels (DCF^+^/PI^+^). DCF^+^-spill over into FL3 detector was compensated (7.3%), and the percentage of viable sperm with high ROS levels (DCF^+^/PI^−^) was corrected using the nonsperm particles from the SYBR-14/PI costaining. For each treatment and incubation time, results are expressed as the percentage of viable sperm with low (DCF^−^/PI^−^) and high ROS levels (DCF^+^/PI^−^) (mean ± SEM; *n* = 9). The intensity of DCF^+^/PI^−^ sperm was determined, the results being expressed as the mean ± SEM (*n* = 9).

### 4.8. Statistical Analyses

Data were analyzed using a statistical package (IBM SPSS Statistics 25.0; Armonk, New York, NY, USA). Distribution of data and homogeneity of variances were tested through Shapiro–Wilk and Levene tests, respectively. A linear mixed model was subsequently run for each sperm parameter. The intrasubject factor was the incubation time (0, 60, 120, 180, 240, 250, 270, and 300 min), and the fixed-effects factor (intersubjects) was the treatment (control and samples blocked with 1, 5, or 10 µM). A post hoc Sidak test was used for pairwise comparisons and the level of significance was set at *p* ≤ 0.05. Data are shown as mean ± SEM.

## 5. Conclusions

The present study confirmed the presence of NHE1 in the plasma membrane of pig sperm, with a molecular weight of 75 kDa. NHE1 located in the connecting and terminal pieces of the flagellum and the equatorial region of the sperm head. Pharmacological blockage of NHE channels has little physiological relevance during in vitro capacitation of mammalian sperm, but it reduces their ability to trigger the acrosome exocytosis and to switch to hyperactivated motility in response to progesterone stimulus. Moreover, NHE channels do not seem to be implicated in the regulation of Ca^2+^ influx during in vitro capacitation in mammalian sperm. Further studies should be focused on the identification of other NHE isoforms in the plasma membrane of mammalian sperm, especially NHE5 and sNHE, and on the description of their specific role in mammalian sperm physiology.

## Figures and Tables

**Figure 1 ijms-22-12646-f001:**
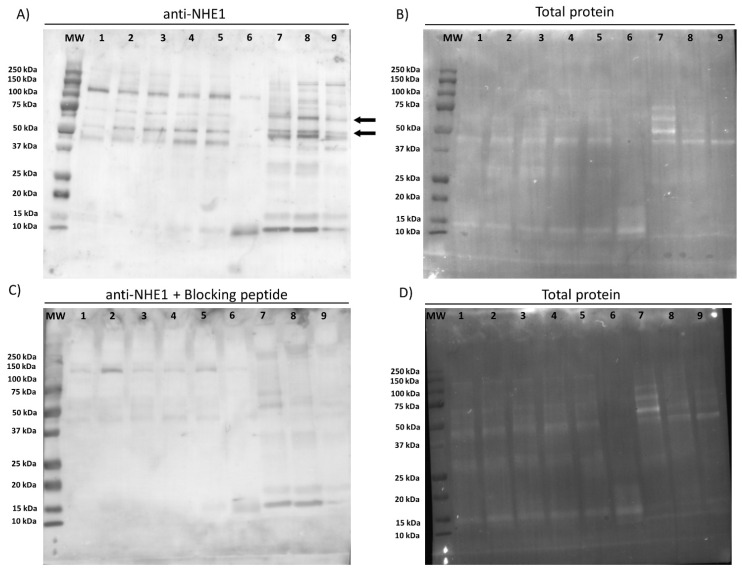
Representative immunoblotting for NHE1 (**A**), incubation of anti-NHE1 with the blocking peptide (**C**), and total protein content (**B**,**D**). Lanes: (MW) protein ladder; (1–6) sperm samples, each one corresponding to a different male; (7) pig ovary tissue sample; (8) pig oviduct tissue sample; (9) pig epididymis tissue sample.

**Figure 2 ijms-22-12646-f002:**
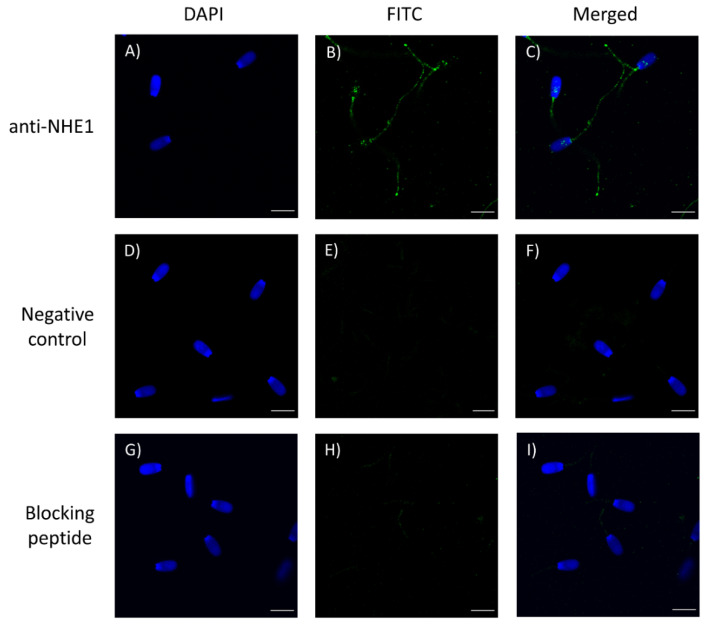
Immunolocalization of NHE1 in the sperm plasma membrane (**A**–**C**), negative control (**D**–**F**), and after the peptide competition assay (**G**–**H**). NHE1 appears stained in green (FITC; fluorescein isothiocyanate) and nuclei are in blue (DAPI; 4′6′-diamidion-2-phenylindole). Scale bar: 9 µm.

**Figure 3 ijms-22-12646-f003:**
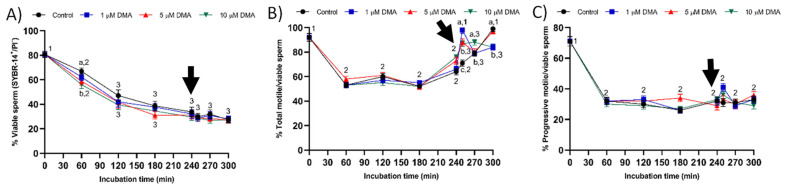
Percentages of viable sperm (**A**), total motile/viable sperm (**B**), and progressively motile/viable sperm (**C**) during in vitro capacitation in the control and samples blocked with 1, 5, and 10 μM DMA. Different letter superscripts indicate significant differences between control and blocked samples in sperm viability and total motility within a single time point (*p* < 0.05); lack of letter superscripts indicate no significant differences in progressive motility between treatments within a single time point (*p* > 0.05). Different numeral superscripts indicate significant differences between two consecutive time points within a treatment (*p* < 0.05). Since the pattern of variation of sperm viability and progressive motility throughout the incubation time was similar between control and blocked samples, all treatments have the same numeral superscript assigned to each time point. The arrow indicates the addition of 10 μg/mL of progesterone at 240 min of incubation. Results are expressed as the mean ± SEM (*n* = 9).

**Figure 4 ijms-22-12646-f004:**
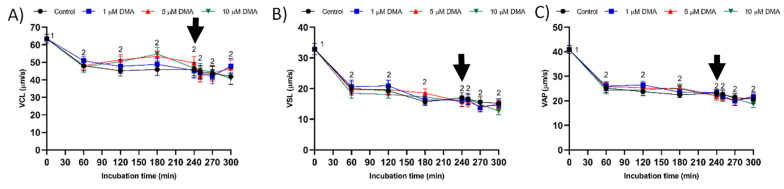
VCL (**A**), VSL (**B**), and VAP (**C**) during in vitro capacitation in the control and samples blocked with 1, 5, and 10 μM of DMA. Different numeral superscripts indicate significant differences between two consecutive time points within a treatment (*p* < 0.05); since the pattern of variation of VCL, VSL, and VAP was similar between control and blocked samples throughout the incubation time, all treatments have the same numeral superscript assigned to each time point. The arrow indicates the addition of 10 μg/mL of progesterone at 240 min of incubation. Results are expressed as the mean ± SEM (*n* = 9).

**Figure 5 ijms-22-12646-f005:**
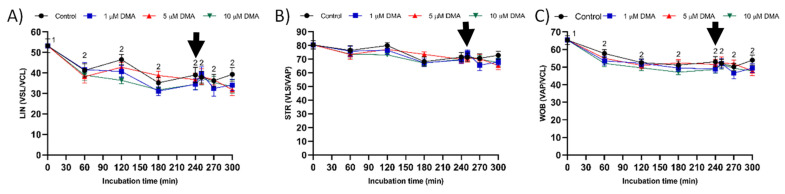
LIN (**A**), STR (**B**), and WOB (**C**) during in vitro capacitation in the control and samples blocked with 1, 5, and 10 μM DMA. Different numeral superscripts indicate significant differences between two consecutive time points within a treatment (*p* < 0.05); since the pattern of variation of LIN and WOB was similar between control and blocked samples throughout the incubation time, all treatments have the same numeral superscript assigned to each time point. Lack of numeral superscripts indicate no significant differences in STR between treatments within a single time point (*p* > 0.05). The arrow indicates the addition of 10 μg/mL of progesterone at 240 min of incubation. Results are expressed as the mean ± SEM (*n* = 9).

**Figure 6 ijms-22-12646-f006:**
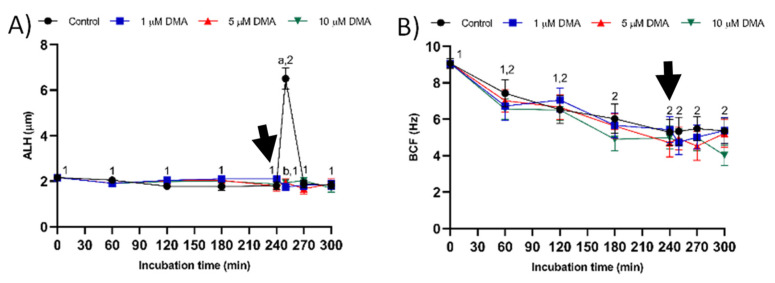
Kinematic parameters ALH (**A**) and BCF (**B**) during in vitro capacitation in the control and samples blocked with 1, 5, and 10 μM DMA. Different letter superscripts indicate significant differences between control and blocked samples within a single time point (*p* < 0.05). Different numeral superscripts indicate significant differences between two consecutive time points within a treatment (*p* < 0.05); since the pattern of variation of BCF throughout the incubation time was similar between control and blocked samples, all treatments have the same numeral superscript assigned to each time point. The arrow indicates the addition of 10 μg/mL of progesterone at 240 min of incubation. Results are expressed as the mean ± SEM (*n* = 9).

**Figure 7 ijms-22-12646-f007:**
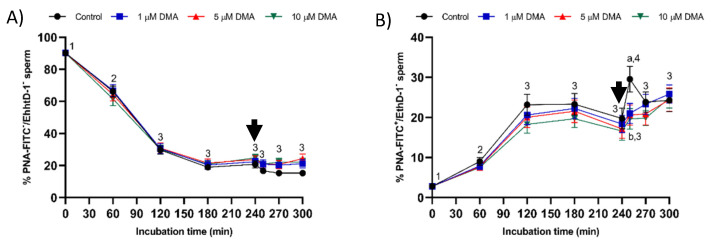
Percentages of viable sperm with an intact acrosome (**A**) and an altered acrosome (**B**) during in vitro capacitation in the control and samples blocked with 1, 5, and 10 μM DMA. Different letter superscripts indicate significant differences between control and blocked samples within a single time point (*p* < 0.05). Different numeral superscripts indicate significant differences between two consecutive time points within a treatment (*p* < 0.05); since the pattern of variation of the percentage of viable sperm with an intact acrosome throughout the incubation time was similar between control and blocked samples, all treatments have the same numeral superscript assigned to each time point. The arrow indicates the addition of 10 μg/mL of progesterone at 240 min of incubation. Results are expressed as the mean ± SEM (*n* = 9).

**Figure 8 ijms-22-12646-f008:**
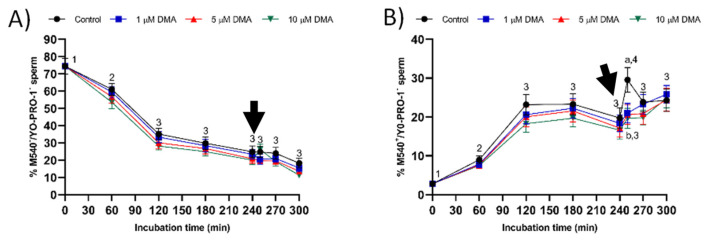
Percentages of viable sperm with low (**A**) and high membrane lipid disorder (**B**) during in vitro capacitation in the control and samples blocked with 1, 5, and 10 μM DMA. Different letter superscripts indicate significant differences between control and blocked samples within a single time point (*p* < 0.05). Different numeral superscripts indicate significant differences between two consecutive time points within a treatment (*p* < 0.05); since the pattern of variation of the percentage of viable sperm with low membrane permeability was similar between control and blocked samples throughout the incubation time, all treatments have the same numeral superscript assigned to each time point. The arrow indicates the addition of 10 μg/mL of progesterone at 240 min of incubation. Results are expressed as the mean ± SEM (*n* = 9).

**Figure 9 ijms-22-12646-f009:**
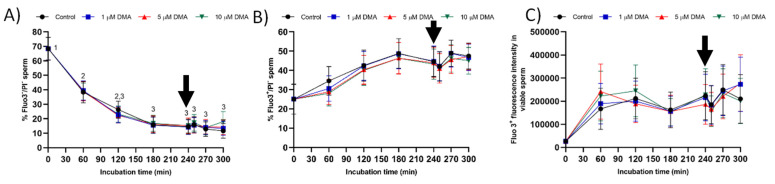
Percentages of viable sperm with low (**A**) and high intracellular calcium levels (**B**), and fluorescence intensity of Fluo3^+^ in viable sperm (**C**) during in vitro capacitation in the control and samples blocked with 1, 5, and 10 μM DMA. Different numeral superscripts indicate significant differences between two consecutive time points within a treatment (*p* < 0.05); since the pattern of variation of the percentage of viable sperm with low intracellular calcium levels was similar between control and blocked samples throughout the incubation time, all treatments have the same numeral superscript assigned to each time point. Lack of numeral superscripts indicate no significant differences in the percentage of viable sperm with high intracellular calcium levels and in Fluo3^+^ fluorescence intensity between treatments within a single time point (*p* > 0.05). The arrow indicates the addition of 10 μg/mL of progesterone at 240 min of incubation. Results are expressed as the mean ± SEM (*n* = 9).

**Figure 10 ijms-22-12646-f010:**
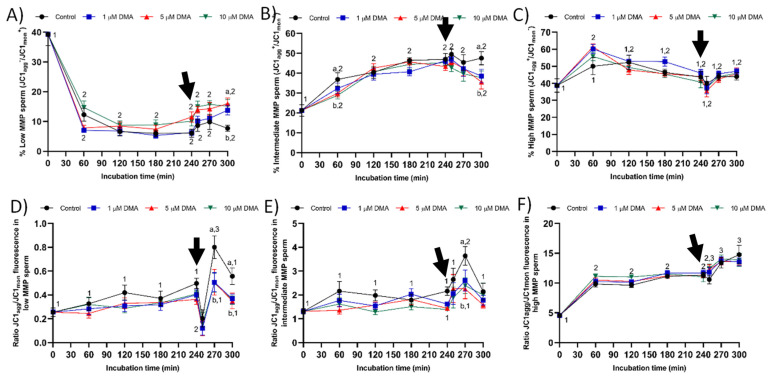
Percentages of sperm with low (**A**), intermediate (**B**), and high (**C**) mitochondrial membrane potential (MMP), and JC1_agg_/JC1_mon_ fluorescence ratio of sperm with low (**D**), intermediate (**E**), and high MMP (**F**) during in vitro capacitation in the control and samples blocked with 1, 5, and 10 μM DMA. Different letter superscripts indicate significant differences between control and blocked samples within a single time point (*p* < 0.05); lack of letter superscripts indicate no significant differences between treatments at a single time point (*p* > 0.05). Different numeral superscripts indicate significant differences between two consecutive time points within a treatment (*p* < 0.05); since the pattern of variation of the percentage of low, intermediate, and high MMP sperm and the ratio JC1_agg_/JC1_mon_ fluorescence in high MMP sperm was similar between control and blocked samples throughout the incubation time, all treatments have the same numeral superscript assigned to each time point. The arrow indicates the addition of 10 μg/mL of progesterone at 240 min of incubation. Results are expressed as the mean ± SEM (*n* = 9).

**Figure 11 ijms-22-12646-f011:**
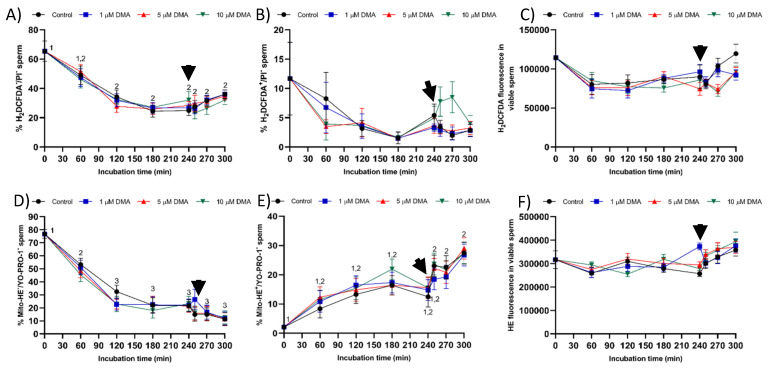
Percentages of sperm with low (**A**,**D**) and high (**B**,**E**) ROS and superoxide levels, and fluorescence intensity of DCF^+^ or E^+^ in viable sperm with high overall ROS (**C**) and superoxide (**F**) levels during in vitro capacitation in the control and samples blocked with 1, 5, and 10 μM DMA. Different numeral superscripts indicate significant differences between two consecutive time points within a treatment (*p* < 0.05); since the pattern of variation of each sperm parameter throughout the incubation time was similar between control and blocked samples, all treatments have the same numeral superscript assigned to each time point. Lack of numeral superscripts indicate no significant differences in the percentage of viable sperm with high ROS levels, and in DCF^+^ and E^+^ fluorescence intensity. The arrow indicates the addition of 10 μg/mL of progesterone at 240 min of incubation. Results are expressed as the mean ± SEM (*n* = 9).

## Data Availability

Data are available from the corresponding upon request.

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
