# Peer review of "Blocking NHE Channels Reduces the Ability of In Vitro Capacitated Mammalian Sperm to Respond to Progesterone Stimulus"

_ijms, 2021, doi:10.3390/ijms222312646_

Round 1
Reviewer 1 Report
Affiliations:
- Line 12: no author is indicated to be affiliated to this institution
- Line 14: no author is indicated to be affiliated to this institution
Abstract:
- Line 33: indicate that you mean before incubation/addition of progesterone
- Line 36: indicate what ALH means
Introduction:
- Line 45-47: rephrase like this: being the Na+/H+ exchangers (NHEs), HCO3- membrane transporters, monocarboxylate transporters (MCTs), and voltage gated H+ transporters (HVCN1) the most studied in mammals.
- Line 60: where are NHA-1 and NHA-2 isoforms found? Are they also localized in the membrane? Please indicate it.
- Line 66: where is the NHE3 isoform found? In which cells/tissues? Please indicate that.
- Line 83: does DMA also inhibit otters NHE subfamilies (NHA-1, NHA-2, sNHE). I think it is very important to clearly indicate that.
- It is very useful for the reader that you introduce the research background, the novelty of your paper in line 75, the objectives of the study in line 79, the methodology in line 81.
- Line 87: here you describe only one of the results of your paper. It would be better not to describe the results, as they are described in the following section or very briefly describe them all, thus adding as well the results from viability, motility, etc.
Results:
- Figure 1: it would be much clearer for the reader if you indicate with an arrow the band that you are referring to in the membranes.
- Figure 2: this figure doesn’t clearly represent the described results. You should use another photo that better suits what you describe or include several photos in the figure that together match what’s described in the text. In particular, in line 108 you state that there’s a weak labelling in the acrosomal region of the sperm head and in line 110 that a diffused immunofluorescent pattern in some cases spreader along the principal piece. However, by carefully looking at the photos, this labelling is not visible. Indeed, the acrosomal labelling is more evident in the photo from the peptide competition assay. You should include a photo or a series of photos where this labelling that you describe is visible.
- Line 119: why does sperm viability also decrease in control group?
- Line 127: please explain for non-experts what does total motile and progressively motile mean.
- Line 129: please replace “maintained” for “remained”
- Line 134: for “significant differences between treatments”, do you mean differences between the treatments and the control or differences between the different treatment groups? Please specify.
- Figure 3: although you already state in the methodology section what the graphics represent (mean, SEM, n), please add it also to the legend of the figures. This applies to figures 3, 4, 5, 6, 7, 8, 9, 10 and 11. Please represent statistics differently, they are not clear and really hard to understand. You indicate that different letter superscripts indicate significant differences between treatment within the same time point. However, there are different letters (a, b, c). What do they mean? The same thing applies to numbers. You indicate that different numeral superscripts indicate significant differences between time points within a treatment. However, there are different numbers (1, 2, 3, 4). What do they mean? Please specify what those letters and numbers mean. For example a = p < 0.05 vs. control, b = p < 0.05 vs. 1 uM DMA, 1 = p < 0.05 vs. 0 min, 2 = p < 0.05 vs. 120 min… Besides, sometimes the superscript are placed over different marks and it gets confusing as it is not clear to which one it makes reference (to the control, to the 1 uM DMA, to the 5 uM DMA, etc).
- Line 146: please specify what VCL, VSL and VAP stand for.
- Line 149: please specify what LIN, WOB and STR stand for. Where written Figure 5A-5C it should be Figure 5A and 5C.
- Line 150: where written Figure 5C it should be Figure 5B.
- Line 160: please specify what ALH stands for.
- Line 164: please specify what BCF stands for.
- Line 180: why is there a peak in the percentage of viable sperm with altered acrosome in the control group at 250 min?
- Figure 7: arrows indicating the addition of progesterone are missing.
- Figure 8B: here as in some other graphics, there is no error bar for any group at time 0 min. Why is that?
- Line 201: does it decrease after 60 min or already from the beginning of the experiment? Does it decrease after 180 min or after 120 min?
- Line 218: the word potential is missing (low mitochondrial membrane potential).
- Line 219: it is mitochondrial membrane potential and not membrane mitochondrial potential.
- Figure 10D: which explanation might there be for the fact that after 250 mins of incubation (=right after adding progesterone) there’s a great decrease in JC1agg/JCmon fluorescence in LMMP? In the case of MMMP and HMMP there’s not such decrease.
- Line 247: H2DCFDA is not a peroxide specific redox probe. Indeed, this probe has showed little specificity and measures as ell hydroxyl, peroxyl and other ROS intermediates.
- Figure 11B and E: the x-axis in these graphics are different from all the others.
Discussion:
- Line 304: if here you’re referring to your results, you should not generalize writing mammalian sperm, but boar sperm.
- Line 315: this is the greatest weakness of this work. Might be negative results due to the fact that DMA might not be blocking sNHE (or other NHE isoforms) and therefore there’s still activity of these exchangers? Please discuss this possibility.
Materials and Methods:
- You have included a very well written and detailed methodology section and it’s clearly visible the effort you made in providing such accurate information.
- Line 422: so I understand reading the text that you made 7 time points: 60, 120, 180, 240, 250 and 270 minutes. What about time point 0 min? Do you have in total 8 time points then?
- Line 432: RIPA lysis buffer (R0278). Was it bought from Sigma? Indicate the provider. Which protease inhibitor did you use?
- Line 616: how do you set the threshold to establish those three groups? (low, intermediate and high MMP).
- Line 639: H2DCFDA is not a hydrogen peroxide specific redox probe. You should bear in mind and also indicate that it is not specific for H2O2 and that, in fact, it is used as a “general ROS” production probe due to its lack of specificity.
- You have explained in this section important information for non-experts in the field as sperm motility and kinematics and all the tests made by flow cytometry. It might be useful for these readers to indicate, maybe at the beginning of the results section, that they should refer to this section for information. Otherwise, readers might think that this information is missing and they find it only when they get reading to the end of the paper.

Author Response
Reviewer 1
- Line 12: no author is indicated to be affiliated to this institution
- Line 14: no author is indicated to be affiliated to this institution
Answer: Thank you very much for your appreciation. In the revised version we corrected this typographical error.
Abstract:
- Line 33: indicate that you mean before incubation/addition of progesterone
Answer: According to your comment, we have clarified this point.
- Line 36: indicate what ALH means
Answer: According to your comment, we have indicated the meaning of ALH.
Introduction:
- Line 45-47: rephrase like this: being the Na+/H+ exchangers (NHEs), HCO3- membrane transporters, monocarboxylate transporters (MCTs), and voltage gated H+ transporters (HVCN1) the most studied in mammals.
Answer: According to your comment, we have rephrased this sentence.
- Line 60: where are NHA-1 and NHA-2 isoforms found? Are they also localized in the membrane? Please indicate it.
Answer: In agreement with your comment, we have added new information related to NHA-1 and NHA-2 isoforms.
- Line 66: where is the NHE3 isoform found? In which cells/tissues? Please indicate that.
Answer: In the original version, we wrote “NHE2-4” isoforms considering that the isoform -3 was included in this rank. Nevertheless, in order to avoid any misunderstanding, we have corrected this sentence to “NHE2, -3 and-4 isoforms”.
- Line 83: does DMA also inhibit otters NHE subfamilies (NHA-1, NHA-2, sNHE). I think it is very important to clearly indicate that.
Answer: After reviewing the related literature, we have found no publication dealing with the inhibitory role of DMA on other NHE subfamilies. Nevertheless, we agree with the reviewer that DMA could also exert a blocking effect on different types of NHE channels. Therefore, we have carefully revised the Introduction and highlight the potential role of DMA on different NHE subfamilies.
- It is very useful for the reader that you introduce the research background, the novelty of your paper in line 75, the objectives of the study in line 79, the methodology in line 81.
Answer: In agreement with your comment, we have carefully revised the content of these lines and highlighted the research background, and the novelty, objectives and methodology of this study.
- Line 87: here you describe only one of the results of your paper. It would be better not to describe the results, as they are described in the following section or very briefly describe them all, thus adding as well the results from viability, motility, etc.
Answer: In agreement with your comment, we have eliminated the sentence referring to the presence and localization of NHE1 channels.
Results:
- Figure 1: it would be much clearer for the reader if you indicate with an arrow the band that you are referring to in the membranes.
Answer: Thank you very much for this suggestion. We have added an arrow to highlight the band.
- Figure 2: this figure doesn’t clearly represent the described results. You should use another photo that better suits what you describe or include several photos in the figure that together match what’s described in the text. In particular, in line 108 you state that there’s a weak labelling in the acrosomal region of the sperm head and in line 110 that a diffused immunofluorescent pattern in some cases spreader along the principal piece. However, by carefully looking at the photos, this labelling is not visible. Indeed, the acrosomal labelling is more evident in the photo from the peptide competition assay. You should include a photo or a series of photos where this labelling that you describe is visible.
Answer: According to your suggestion, we have carefully revised this figure and improved the quality of images.
- Line 119: why does sperm viability also decrease in control group?
Answer: This is a current effect extensively reported in nearly all the experiments on sperm capacitation. Nevertheless, and following your comment, we have added a sentence highlighting the effects of in vitro capacitation on control samples, which agreed with previous studies.
- Line 127: please explain for non-experts what does total motile and progressively motile mean.
Answer: We have clarified this point by adding a brief definition of total and progressively motility.
- Line 129: please replace “maintained” for “remained”
Answer: We did this change.
- Line 134: for “significant differences between treatments”, do you mean differences between the treatments and the control or differences between the different treatment groups? Please specify.
Answer: We have clarified this point by indicating that we observed significant differences between blocked and control samples.
- Figure 3: although you already state in the methodology section what the graphics represent (mean, SEM, n), please add it also to the legend of the figures. This applies to figures 3, 4, 5, 6, 7, 8, 9, 10 and 11. Please represent statistics differently, they are not clear and really hard to understand. You indicate that different letter superscripts indicate significant differences between treatment within the same time point. However, there are different letters (a, b, c). What do they mean? The same thing applies to numbers. You indicate that different numeral superscripts indicate significant differences between time points within a treatment. However, there are different numbers (1, 2, 3, 4). What do they mean? Please specify what those letters and numbers mean. For example a = p < 0.05 vs. control, b = p < 0.05 vs. 1 uM DMA, 1 = p < 0.05 vs. 0 min, 2 = p < 0.05 vs. 120 min… Besides, sometimes the superscript are placed over different marks and it gets confusing as it is not clear to which one it makes reference (to the control, to the 1 uM DMA, to the 5 uM DMA, etc).
Answer: The notation used to highlight the differences between treatments within a given time point, and between points within a given treatment is widely used in most manuscripts. In fact, we have published several papers using both letter and numeral superscripts in Tables and Figures. Actually, it is difficult to find another way to depict differences between treatments and along time. In spite of this, we are willing to accept, along with the reviewer, that these superscripts may be sometimes hard to understand. For this reason, and in order to satisfy the reviewer, we have carefully revised all figure legends to provide a more accurate information about the meaning of superscripts.
- Line 146: please specify what VCL, VSL and VAP stand for.
Answer: We have added the meaning of these terms.
- Line 149: please specify what LIN, WOB and STR stand for. Where written Figure 5A-5C it should be Figure 5A and 5C.
Answer: We have added the meaning of these terms and corrected the typographical error.
- Line 150: where written Figure 5C it should be Figure 5B.
Answer: We have corrected this typographical error.
- Line 160: please specify what ALH stands for.
Answer: We have added the meaning of this term.
- Line 164: please specify what BCF stands for.
Answer: We have added the meaning of this term.
- Line 180: why is there a peak in the percentage of viable sperm with altered acrosome in the control group at 250 min?
Answer: The peaks indicates that in control samples progesterone addition activates the sequence of events resulting in acrosomal exocytosis. In contrast, in samples incubated with DMA this process is blocked, regardless of DMA concentration. Nevertheless, and to improve the understanding of our manuscript, we have carefully revised this subsection and specified that in control samples this peak corresponds to acrosomal exocytosis, whereas in blocked acrosomal exocytosis was impaired.
- Figure 7: arrows indicating the addition of progesterone are missing.
Answer: Thank you very much for your comment. We have corrected this error.
- Figure 8B: here as in some other graphics, there is no error bar for any group at time 0 min. Why is that?
Answer: All variables have the error bar at time 0, but in some graphics, it is difficult to see because its low value. For instance, in graphic 8B mean ± SEM at time 0 is 2.82 ± 0.38. Since SEM value is very low, it is very difficult to appreciate in the plot. We have carefully reviewed all the plots to check that the mean ± SEM of all values was accurately represented, despite SEM being difficult to appreciate when it is < 1.
- Line 201: does it decrease after 60 min or already from the beginning of the experiment? Does it decrease after 180 min or after 120 min?
Answer: In order to provide a clear message, we have carefully revised this sentence and rewritten the results.
- Line 218: the word potential is missing (low mitochondrial membrane potential).
Answer: Thank you very much for your comment. We have added this word in the text.
- Line 219: it is mitochondrial membrane potential and not membrane mitochondrial potential.
Answer: We have corrected this grammatical error.
- Figure 10D: which explanation might there be for the fact that after 250 mins of incubation (=right after adding progesterone) there’s a great decrease in JC1agg/JCmon fluorescence in LMMP? In the case of MMMP and HMMP there’s not such decrease.
Answer: We agree that the results obtained are quite surprising. Please, note that in the Discussion section we focused extensively on this point and highlighted the need of future studies to better understand the functional relationship between NHE channels and mitochondrial activity.
- Line 247: H2DCFDA is not a peroxide specific redox probe. Indeed, this probe has showed little specificity and measures as ell hydroxyl, peroxyl and other ROS intermediates.
Answer: To be honest, we are not expert enough on this matter and we followed previously published material where this probe was stated to be specific for hydrogen peroxides (Guthrie and Welch, 2006; Galluzzi et al. Conceptual background and bioenergetic/mitochondrial aspects of oncometabolism, 2014). However, the reviewer may be right that the probe is not a specific peroxide one. For this reason, we have revised the Manuscript and we now refer to its ability to detect overall ROS levels rather than those of hydrogen peroxide.
- Figure 11B and E: the x-axis in these graphics are different from all the others.
Answer: Thank you very much for your appreciation. We have corrected these two graphs.
Discussion:
- Line 304: if here you’re referring to your results, you should not generalize writing mammalian sperm, but boar sperm.
Answer: In agreement with your comment, we have changed “mammalian” to “pig”.
- Line 315: this is the greatest weakness of this work. Might be negative results due to the fact that DMA might not be blocking sNHE (or other NHE isoforms) and therefore there’s still activity of these exchangers? Please discuss this possibility.
Answer: We agree that this could be a reason for these results, so that we have extensively discussed this possibility in the Discussion section.
Materials and Methods:
- You have included a very well written and detailed methodology section and it’s clearly visible the effort you made in providing such accurate information.
Answer: Thank you very much; we sincerely appreciate your comment.
- Line 422: so I understand reading the text that you made 7 time points: 60, 120, 180, 240, 250 and 270 minutes. What about time point 0 min? Do you have in total 8 time points then?
Answer: Please, note that we analysed 8 points: 0, 60, 120, 180, 240, 250, 270 and 300 minutes. Nevertheless, in order to avoid confusions, we have revised and rewritten these lines.
- Line 432: RIPA lysis buffer (R0278). Was it bought from Sigma? Indicate the provider. Which protease inhibitor did you use?
Answer: Please, note that at the beginning of the Material and Methods section we specify that all chemicals were purchased from Sigma-Aldrich unless otherwise indicated. Besides, we have also specified that we used a commercial protease inhibitor and provided the reference.
- Line 616: how do you set the threshold to establish those three groups? (low, intermediate and high MMP).
Answer: Thank you very much for your comment. The three populations were identified based on the presence/absence of JC1-monomers and JC1-aggregates, as follows: (1) sperm with mitochondria presenting low MMP (low MMP; JC-1mon; green-stained); (2) sperm with mitochondria showing intermediate MMP (JC-1agg; orange-stained); and (3) sperm with heterogeneous mitochondria (green and orange). This has been clarified in the subsection about the protocol in M&M.
- Line 639: H2DCFDA is not a hydrogen peroxide specific redox probe. You should bear in mind and also indicate that it is not specific for H2O2 and that, in fact, it is used as a “general ROS” production probe due to its lack of specificity.
Answer: As indicated in our response to a previous comment, we have revised the text and we now refer to its ability to detect overall ROS levels rather than those of hydrogen peroxide.
- You have explained in this section important information for non-experts in the field as sperm motility and kinematics and all the tests made by flow cytometry. It might be useful for these readers to indicate, maybe at the beginning of the results section, that they should refer to this section for information. Otherwise, readers might think that this information is missing and they find it only when they get reading to the end of the paper.
Answer: We completely agree with your comment, but this is a current problem when the Material and Methods section is placed at the end of the manuscript. In spite of this, we do not think that repeating the information in two different sections of the manuscript would be appropriate. Please, note that we just follow the editorial roles.
Reviewer 2 Report
Yeste et al. have reported the presence and functional relevance of the Na+/H+ exchanger NHE1 in pig sperm in a detailed manner. Although the data does not show very drastic effect of blocking NHE1 by DMA, it clearly depics inter-species variation among mammalian sperm. I have a few suggestions, which can be addressed for clarifying the message to the readers:
- What is the predicted molecular weight of pig NHE-1? A clarification mentioning that the higher molecular weight might correspond to glycosylation while smaller bands might have arisen due to degradation of protein during sample preparation need to be added. What is the explanation for the absence of 120kDa bands upon peptide blocking in positive controls, but not in the sperm samples?
- An interesing observation from the peptide block immunofluorescence images is very specific NHE-1 signal at the acrosome. This acrosomal localization is not seen in absence of peptide blocking. Hence, if this is not a general occurence across all the individuals tested, a more representative image should be provided at the peptide control.
- DMA induces the total blockage of NHE1 83 and NHE2 isoforms and partial blockage of NHE5 isoforms. What is the expression level and localization of NHE2 and NHE5 in pig sperm? This needs to be clarified at the introduction section itself to indicate the specificity of DMA in pig sperm. Else a disclaimer may be added in discussion and conclusion that partial effect on DHE2 and DHE5 cannot be ruled out.
- TableS1, VideoS1 and FigureS1 are not mentioned anywhere in main text, they appear only at the Supplementary mat section. It is not even available to the reviewers.
- At the last line of each figure legend the word "progesterone" is missing.
- The readers will be clueless as to what the numerical or alphabetical P-values exactly correspond to interms of P-value. With what parameter are they compared. This has to be clarified.
- There almost no difference among different conc. of DMA treatment. Won't it be better to show the effect on viability with different doses but show only one dose of DMA in all subsequent graphs?
- I suggest toning down the claim to NHE1 being "essential" for triggering acrosome exocytosis and hypermotility after progesterone stimulus in the abstract.
Author Response
Reviewer 2
Yeste et al. have reported the presence and functional relevance of the Na+/H+ exchanger NHE1 in pig sperm in a detailed manner. Although the data does not show very drastic effect of blocking NHE1 by DMA, it clearly depics inter-species variation among mammalian sperm. I have a few suggestions, which can be addressed for clarifying the message to the readers:
- What is the predicted molecular weight of pig NHE-1? A clarification mentioning that the higher molecular weight might correspond to glycosylation while smaller bands might have arisen due to degradation of protein during sample preparation need to be added. What is the explanation for the absence of 120kDa bands upon peptide blocking in positive controls, but not in the sperm samples?
Answer: As extensively indicated in both Introduction and Discussion section, little data exist on NHE1 channels in mammalian sperm and the molecular weight could differ between species and even between tissues from the same species. Please, note that in the revised version this point has been discussed extensively.
- An interesing observation from the peptide block immunofluorescence images is very specific NHE-1 signal at the acrosome. This acrosomal localization is not seen in absence of peptide blocking. Hence, if this is not a general occurence across all the individuals tested, a more representative image should be provided at the peptide control.
Answer: In agreement with your comment, we have carefully revised all the figures and improved the quality of images.
- DMA induces the total blockage of NHE1 83 and NHE2 isoforms and partial blockage of NHE5 isoforms. What is the expression level and localization of NHE2 and NHE5 in pig sperm? This needs to be clarified at the introduction section itself to indicate the specificity of DMA in pig sperm. Else a disclaimer may be added in discussion and conclusion that partial effect on DHE2 and DHE5 cannot be ruled out.
Answer: Please note that both the Introduction and Discussion indicate that little data about the NHE content in mammalian sperm exist, most studies having been performed in mice and only few studies having focused on the NHE content in humans and rams. According to these studies, sperm cells mainly have NHE1, NHE5 and sNHE isoforms. Despite being already described in the Introduction section and following your comment, we have revised both the Introduction and Discussion sections to clarify these points.
- TableS1, VideoS1 and FigureS1 are not mentioned anywhere in main text, they appear only at the Supplementary mat section. It is not even available to the reviewers.
Answer: Please note that we did not upload any supplementary Table and Figure, or any video.
- At the last line of each figure legend the word "progesterone" is missing.
Answer: We apologize for this typographical error. We have added this word in all figure legends.
- The readers will be clueless as to what the numerical or alphabetical P-values exactly correspond to interms of P-value. With what parameter are they compared. This has to be clarified.
Answer: We would like to note that the assignment of different letter and numeral superscripts is the current way to highlight the statistical differences between treatments and temporal points. Moreover, we have published several manuscripts using these superscripts. Nevertheless, according to your comment, we have carefully revised all figure legends in order to provide a clear message.
- There almost no difference among different conc. of DMA treatment. Won't it be better to show the effect on viability with different doses but show only one dose of DMA in all subsequent graphs?
Answer: Indeed, in this experiment we did not observe differences between DMA concentrations. However, we think that this is an important result of our study since it indicates that other channels may also be implicated in the regulation of sperm physiology during in vitro capacitation and after progesterone-induced acrosomal exocytosis. Nevertheless, and as per the reviewer’s request, we have highlighted this lack of differences between DMA concentrations, which we honestly think are better perceived by the reader if all, and not only one, DMA doses are shown.
- I suggest toning down the claim to NHE1 being "essential" for triggering acrosome exocytosis and hypermotility after progesterone stimulus in the abstract.
Answer: In agreement with your comment, we have revised this sentence.